# Efficacy of Fully Covered Self-Expandable Metal Stents for Distal Biliary Obstruction Caused by Pancreatic Ductal Adenocarcinoma: Primary Metal Stent vs. Metal Stent following Plastic Stent

**DOI:** 10.3390/cancers15113001

**Published:** 2023-05-31

**Authors:** Chi-Huan Wu, Sheng-Fu Wang, Mu-Hsien Lee, Yung-Kuan Tsou, Cheng-Hui Lin, Li-Ling Chang, Kai-Feng Sung, Nai-Jen Liu

**Affiliations:** 1Department of Gastroenterology and Hepatology, Linkou Chang Gung Memorial Hospital, Taoyuan 33305, Taiwan; s19821737@gmail.com (C.-H.W.); shanelily@msn.com (S.-F.W.); r5266@adm.cgmh.org.tw (M.-H.L.); flying@adm.cgmh.org.tw (Y.-K.T.); linchehui@adm.cgmh.org.tw (C.-H.L.); h12153@adm.cgmh.org.tw (K.-F.S.); 2Department of Nursing, Linkou Chang Gung Memorial Hospital, Taoyuan 33305, Taiwan; lilian98@cgmh.org.tw

**Keywords:** pancreatic ductal adenocarcinoma, fully covered self-expandable metal stent, recurrent biliary obstruction

## Abstract

**Simple Summary:**

Pancreatic ductal adenocarcinoma can cause distal bile duct obstruction. Fully covered biliary self-expandable metal stents (FCSEMSs) are widely used for bile duct drainage to prevent tissue ingrowth and can be easily exchanged. This retrospective study aimed to compare the efficacy of FCSEMSs placed in a first session of endoscopic retrograde cholangiopancreatography with that of FCSEMSs placed following a prior plastic stent. We also assessed the risk factors for recurrent biliary obstruction of FCSEMSs. We found that the time to recurrent biliary obstruction was comparable between the primary use of FCSEMSs and FCSEMSs with prior plastic stents. We also found that FCSEMS length was associated with dysfunction. These findings provide clinical evidence for the use of FCSEMSs in patients with pancreatic ductal adenocarcinoma and malignant distal obstruction.

**Abstract:**

Fully covered self-expandable metallic stents (FCSEMSs) are inserted in patients with unresectable pancreatic ductal adenocarcinoma (PDAC) to resolve malignant distal bile duct obstructions. Some patients receive FCSEMSs during primary endoscopic retrograde cholangiopancreatography (ERCP), and others receive FCSEMSs during a later session, after the placement of a plastic stent. We aimed to evaluate the efficacy of FCSEMSs for primary use or following plastic stent placement. A total of 159 patients with pancreatic adenocarcinoma (m:f, 102:57) who had achieved clinical success underwent ERCP with the placement of FCSEMSs for palliation of obstructive jaundice. One-hundred and three patients had received FCSEMSs in a first ERCP, and 56 had received FCSEMSs after prior plastic stenting. Twenty-two patients in the primary metal stent group and 18 in the prior plastic stent group had recurrent biliary obstruction (RBO). The RBO rates and self-expandable metal stent patency duration did not differ between the two groups. An FCSEMS longer than 6 cm was identified as a risk factor for RBO in patients with PDAC. Thus, choosing an appropriate FCSEMS length is an important factor in preventing FCSEMS dysfunction in patients with PDAC with malignant distal bile-duct obstruction.

## 1. Introduction

The pancreas is located deep inside the abdomen and surrounded by other organs, making it hard to detect the presence of a neoplasm there. Among all kinds of pancreatic neoplasm, pancreatic ductal adenocarcinoma (PDAC) is one of the most fatal malignancies worldwide. According to the American Cancer Society, in 2023, it is estimated that approximately 64,050 people in the United States will be diagnosed with PDAC, and approximately 50,550 will die from the disease. In addition, PDAC can be difficult to detect and diagnose in its early stages [1]. The 5-year survival rate at the time of diagnosis is 10%, as approximately 80–85% of patients present with either unresectable or metastatic disease [2]. Except for the small number of patients eligible for surgical resection, those with PDAC are susceptible to obstructive jaundice, which occurs when the tumor blocks the distal common bile duct. For these patients, further treatment before palliation of bile-duct obstruction is difficult.

Endoscopic retrograde cholangiopancreatography (ERCP) with the placement of bile-duct self-expandable metal stents (SEMSs) is an effective and minimally invasive method of bile-duct decompression in patients with PDAC [3,4]. Among all types of SEMSs, the initial placement of covered biliary SEMSs (compared with replacement of uncovered biliary SEMSs) can reduce the risk of recurrent stent obstruction due to tumor ingrowth [5,6,7]. Recently, fully covered self-expandable metallic stents (FCSEMSs) have become more commonly used for PDAC-related obstructive jaundice because of their increased biliary patency, which results from improved prevention of tumor ingrowth and tissue hyperplasia [8,9,10,11]. In PDAC patients with FCSEMSs, some receive FCSEMSs for the first time during ERCP, and the remainder receive FCSEMSs after the biliary placement of plastic stents. However, the clinical outcomes of patients receiving a primary or FCSEMS after plastic stents are unclear. We aimed to demonstrate the efficacy of first-use FCSEMSs compared with FCSEMSs used following biliary plastic stenting in patients with PDAC with malignant distal bile-duct obstruction.

## 2. Materials and Methods

### 2.1. Institutional Review Board Statement

This single-center retrospective study was approved by the IRB of our institution. Data acquisition and analysis were performed in accordance with institutional guidelines and regulations. Owing to the retrospective design of the study, the requirement for informed consent was waived by the ethics committee.

### 2.2. Data Collection

We included patients with PDAC who underwent FCSEMS placement for malignant distal bile-duct obstruction between January 2016 and December 2021. The diagnosis of pancreatic ductal adenocarcinoma was based on pathological and/or typical radiological findings. Clinical success was defined as a decline in serum bilirubin concentration of more than 50% or a return to the normal range within 2 weeks of stent placement [12]. Recurrent biliary obstruction (RBO) was defined as an initial clinical success with biliary stenting, but with typical symptoms of bile-duct obstruction, such as fever and abnormal liver function test results, during follow-up. We excluded patients without clinical success after FCSEMS placement.

The primary metal stent (PMS) group included patients receiving an FCSEMS for their first stent. In contrast, the prior plastic stent (PPS) group was defined as those receiving an FCSEMS following the removal of a previous biliary plastic stent. All data, including endoscopic and radiological reports, were collected from the patient medical records. These included patient and tumor characteristics, post-stenting complications such as post ERCP pancreatitis and cholecystitis, and post-FCSEMS treatments such as chemotherapy and radiotherapy. The diagnosis of cholangitis was based on findings of acute inflammation, laboratory data, and imaging, following the 2018 Tokyo Guidelines [13]. The outcomes were the rates of RBO and stent patency in both groups. The time to RBO was defined as the period between stent placement and biliary stent dysfunction, or between stent placement and death if RBO was observed in a postmortem examination.

### 2.3. Endoscopic Biliary Drainage Procedure

All FCSEMSs were deployed across the papilla of Vater. ERCP was performed for retrograde stenting for the placement of FCSEMSs. The endoscopic procedures were performed using a standard duodenoscope (TJF 260 or JF 260; Olympus Optical Co., Ltd., Tokyo, Japan) under prone position. The procedure was performed under moderate sedation with intravenous midazolam and fentanyl. The length of the bile duct stricture was determined by comparing it with the width of the duodenoscope under direct fluoroscopy after contrast medium injection. In the PMS group, fully covered biliary SEMSs, 10 mm in diameter (BONASTENT; Standard Sci-Tech Inc., Seoul, Korea), were placed after a small-to-moderate endoscopic sphincterotomy during the first ERCP. The metal wire of the FCSMES used in our study consists of nitinol with a silicone-covered membrane on the entire stent. The features of this FCSEMS include a woven hook-and-cross structure, which produces high radial force and low axial force [14]. The FCSEMS extended at least 1 or 2 cm above the top of the stricture. The distal end of the FCSEMS was placed in the duodenal lumen and protruded from the papilla by approximately 1 cm.

In the PPS group, 10-Fr. plastic stents (Advanix, Boston Scientific, Marlborough, MA, USA) 5 cm or 7 cm long were placed during a previous ERCP following small- to moderate-sized sphincterotomy. We exchanged the plastic stents with FCSEMSs during the next session of ERCP, usually after 3 months. We brought the time of the stent exchange forward if cholangitis or recurrent bile duct obstruction was encountered. After plastic stent removal, the bile duct was cleaned to the extent possible using a stone-retrieval balloon or basket, followed by the placement of the FCSEMS (BONASTENT; 10 mm in diameter).

### 2.4. Statistical Analysis

Categorical variables are presented as frequencies and percentages. Continuous variables are presented as medians and ranges. Categorical variables were analyzed using Fisher’s exact test, and *t*-tests were used for continuous variables. Potential risk factors for RBO were analyzed initially using univariable Cox regression analysis. These factors were characterized by hazard ratios (HRs) with 95% confidence intervals (CIs). Variables with a *p*-value of less than 0.25 were included in a subsequent multivariable Cox regression analysis to estimate an adjusted HR with 95% CIs. The time to RBO was estimated using the Kaplan–Meier method and compared between groups using the log-rank test. Statistical analyses were performed using IBM SPSS Statistics version 25 (IBM, Armonk, NY, USA). Statistical significance was set at a two-sided *p*-value < 0.05.

## 3. Results

### 3.1. Patient Characteristics

In total, 103 patients with PDAC had primary FCSEMSs placed during a first ERCP for distal bile-duct obstruction, and 56 patients had FCSEMS placement and PPS use. The baseline characteristics of the patients in the PMS and PPS groups are shown in Table 1. The two groups were similar in age, sex, and tumor stage (including liver metastasis, ascites, and duodenal obstruction). Total bilirubin, alanine aminotransferase (ALT), and alkaline phosphatase (ALP) before ERCP were much higher in the PMS group. However, cholangitis before FCSEMS placement was much higher in the PPS group (*p* < 0.001). Tumor markers for PDAC, such as carcinoembryonic antigen (CEA) and carbohydrate antigen-199 (CA-199), were not significantly different between the two groups. There was also no significant difference between the groups regarding anti-cancer treatments, including chemotherapy or radiotherapy, following FCSEMS placement.

### 3.2. ERCP Procedure and FCSEMS Placement

We observed no significant intergroup difference in the length of the biliary stricture (22 vs. 20 mm, *p* = 0.900). The length of the FCSEMS ranged from 6 to 8 cm; 6-cm-long FCSEMSs were used mostly for biliary drainage. Regarding post-ERCP complications, the incidence of pancreatitis was comparable between the two groups (2.91% vs. 3.57%, *p* = 1.000). Post-SEMS cholecystitis, which reportedly has a higher rate with covered SEMSs [15], showed a higher rate in the PMS group than in the PPS group, but the difference was not significant (7.76% vs. 3.57%, *p* = 0.476).

### 3.3. FCSEMS Dysfunction

During the follow-up period, 22 and 18 patients in the PMS and PPS groups, respectively, developed RBO. The cumulative incidence of RBO was similar (PMS, 21.25% vs. PPS, 32.14%, *p* = 0.180). Major causes of occlusion in the PMS group were the presence of sludge or food impaction (40.91%), tumor overgrowth (31.82%), tumor ingrowth at follow-up (13.63%), and stent migration (13.63%); in the PPS group, sludge or food impaction (55.56%), tumor overgrowth (22.22%), tumor ingrowth (11.11%), and stent migration (11.11%). In each cause of RBO, the incidence was not significantly different between the two groups. The median days to RBO was 152 (95% CI, 132–213) in the PMS group and 177.5 (95% CI, 145–228) in the PPS group, indicating no significant intergroup difference (*p* = 0.529).

### 3.4. Risk Factors for FCSEMS Dysfunction in PDAC Patients with Malignant Distal Bile-Duct Obstruction

Table 2 shows the results of analyses to identify the risk factors for FCSEMS dysfunction in patients with PDAC and malignant distal bile-duct obstruction. Known potential risk factors include prior plastic stent drainage [16], bilirubin level [16,17], cholangitis before FCSEMS placement [18], tumor size [19], length of bile-duct stricture [19], duodenal invasion [19], liver metastasis [19], ascites [19], and length of the FCSEMS [20]. These potential risk factors were further analyzed using Cox regression analysis. Multivariable analysis using the Cox regression model showed that SEMS length > 6 cm was an independent predicting factor for FCSEMS dysfunction (HR 0.631, 95% confidence interval [CI 0.414–0.691, *p* = 0.032). We used a Kaplan–Meier curve to estimate the cumulative time to FCSEMS dysfunction. The median time to FCSEMS dysfunction was significantly shorter in PDAC patients with SEMS length >6 cm than in PDAC patients with 6-cm FCSEMSs (141 vs. 170 days, *p* = 0.014) (Figure 1). These factors also showed significant differences in the log-rank test.

## 4. Discussion

Endoscopic transpapillary stenting is a widely accepted treatment as the first-line approach to biliary drainage in PDAC patients with malignant distal biliary obstruction. Plastic stents are easy to insert, remove or exchange; however, these plastic stents are associated with a statistically significantly higher recurrent occlusion, higher treatment failure, higher need for reintervention, and higher possibility of cholangitis compared with SEMSs [21]. SEMSs could serve a longer patency duration than that of plastic stents. [16,22]. According to the guidelines of the European Society of Gastrointestinal Endoscopy, SEMSs have advantages over plastic stents in terms of the longer survival of patients, lower occlusion and cholangitis risk of stents, and lower need for reintervention [23]. Therefore, SEMSs are preferred to plastic stents for the palliative treatment of malignant distal bile-duct obstruction in patients with unresectable PDAC. There are three types of SEMS: partially covered SEMS, fully covered SEMS, and uncovered SEMS. To overcome the drawbacks of plastic stents, uncovered SEMSs were introduced in the 1980s and have a longer patency due to their larger diameter. However, one of the reasons for stent dysfunction with uncovered SEMSs is tumor ingrowth, which does not happen with plastic stents [24,25]. Partially covered SEMSs have been developed, which have a silicone-covered membrane at the central part of the stent to prevent tumor ingrowth. Previous randomized controlled trials from Japan have shown that partially covered SEMSs provide a longer patency period than uncovered SEMS due to their advantage of preventing tumor ingrowth [26]. FCSEMSs were initially designed to overcome benign biliary stricture, such as chronic pancreatitis related biliary stricture, post operative benign biliary stricture, or post liver transplant biliary anastomosis stricture [27,28,29]. Unlike partially covered SEMSs, FCSEMSs do not have 5-mm uncovered flared portions at both ends, which means the silicone membrane covers the entire stent. FCSEMSs are easy to remove after the benign bile duct stricture disappears, just like plastic stents. For patients with malignant biliary stricture, FCSEMSs have better efficacy compared with partially covered or uncovered SEMSs because they have a coating on the entire metal stent that prevents tumor ingrowth and advantageously permits SEMS removal for exchange during ERCP [30]. Despite the benefits of FCSEMSs, some patients with PDAC choose to undergo plastic stenting first for various reasons, notably the high cost of SEMSs. Our results show that the RBO rate and patency period of FCSEMSs were comparable between the PMS and PPS groups. The result was the same in the multivariate analysis, which revealed that PPS placement was not a risk factor for RBO. A previous study on PDAC patients with partially covered SEMSs found that prior plastic stenting was the only significant factor for RBO in a univariable analysis, but it was eliminated during multivariable analysis [31].

The rate of cholangitis before SEMS placement was much higher in the PPS group, but this factor did not influence the RBO rate or FCSEMS patency. In previous studies discussing the RBO of uncovered SEMSs, cholangitis before SEMS placement played a significant role in the development of RBO [18]. One possible explanation for cholangitis having different associations for uncovered SEMSs vs. FCSEMSs is that bacteria adhere to and subsequently form a biofilm on the ragged bile-duct mucosal surface caused by tumor ingrowth, leading to the aggregation of bile sludge and thus RBO [32]. FCSEMSs provide a smooth surface owing to the presence of a silicone-covered membrane in the bile duct that protects against both tumor ingrowth and bacteria-induced biofilm formation in all stented areas. Based on our results, we recommend that FCSEMSs be introduced first to patients with PDAC requiring an SEMS for bile drainage who have previously undergone biliary plastic stenting with a high risk of cholangitis.

We also identified the risk factors for RBO. Most PDAC-related bile-duct obstructions occur in the distal duct; a 6-cm FCSEMS is usually long enough for these patients and long enough to ensure that the distal end protrudes from the papilla by approximately 1 cm. However, the proximal end of a long (>6 cm) FCSEMS is close to the hilar area. If tissue overgrowth or bile sludge occurs in patients with long SEMSs, the site of obstruction may be beyond the proximal part of the SEMS, which is very close to the hepatic hilum (Figure 2). When obstruction occurs close to the hilum, even relatively small tissue hyperplasia or sludge accumulation can cause clinical symptoms compared with obstruction in the distal part of the common bile duct. In our study, long SEMSs (>6 cm) also had a greater risk of RBO than did short SEMSs (6 cm). In a previous study that included patients with unresectable and borderline-resectable PDAC, long SEMSs (>6 cm) also had a short time to RBO, but this association did not reach statistical significance in a multivariable analysis [20]. The reason may be that a borderline-resectable case is a risk-reducing factor for RBO because of the limited observation period due to some of the patients’ wounds being referred to a surgeon for pancreaticoduodenectomy before RBO. Another study that considered all types of malignancy-related distal bile duct obstructions also found that long FCSEMSs had a short period to RBO [33]. In our study, all patients had unresectable PDACs, and the difference between short and long SEMSs was more pronounced during the long-term follow-up. These results suggest that choosing an appropriate length of FCSEMS to maintain some distance from the hepatic hilum is advisable as this may prolong stent patency and avoid RBO.

This study had several limitations. First, this was a retrospective study with an unequal sample size in the comparison groups. Second, some patients in the PPS group switched to an FCSEMS during a scheduled stent change during ERCP and had no cholangitis or hyperbilirubinemia. Third, our study lacked data on the patient responses to anti-cancer treatment. Further studies based on prospectively collected data are needed to determine the best time to place FCSEMSs to relieve malignant distal bile duct obstruction in PDAC patients.

## 5. Conclusions

We demonstrated that FCSEMSs could provide equal efficacy in releasing malignant distal bile duct obstruction in patients with PDAC, whether used primarily or after plastic stent placement. An FCSEMS length greater than 6 cm was identified as a risk factor for RBO after FCSEMS placement in patients with PDAC. Further prospective studies are required to validate these findings.

## Figures and Tables

**Figure 1 cancers-15-03001-f001:**
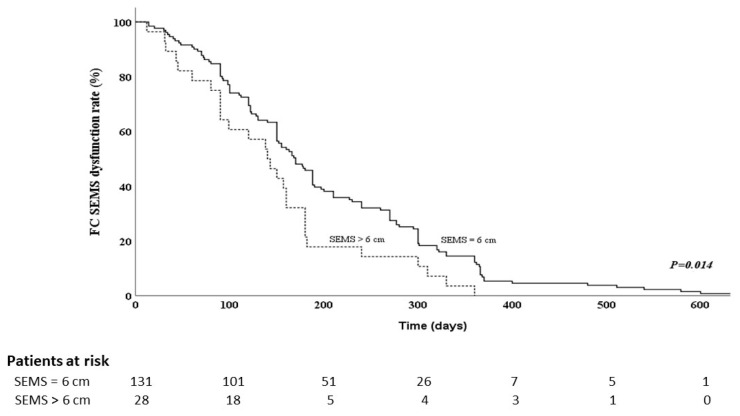
Kaplan–Meier curve for fully covered self-expandable metal stents (SEMSs), showing time to dysfunction vs. length of stent.

**Figure 2 cancers-15-03001-f002:**
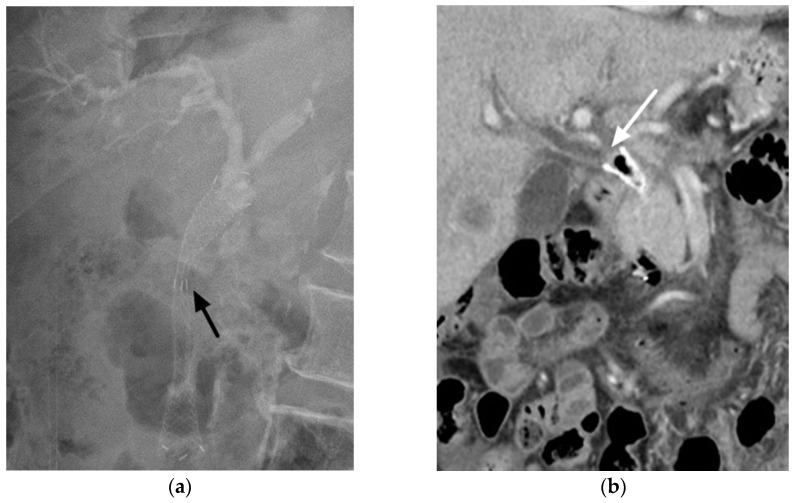
Images from a case of a 72-year-old man with pancreatic ductal adenocarcinoma with malignant bile-duct obstruction. (**a**) One 7-cm fully covered biliary self-expandable metal stent was inserted to relieve symptoms during the first endoscopic retrograde cholangiopancreatography. Radiograph showing the narrowed part of the bile duct at the distal end (black arrow). The proximal end of the metal stent is directly below the hepatic hilum. (**b**) Approximately 8 months after metal stent placement, the patient experienced recurrent biliary duct obstruction with fever and jaundice. Computed tomography revealing tissue overgrowth beyond the proximal end of the metal stent that involves the bifurcation of the right and left hepatic ducts (white arrow).

**Table 1 cancers-15-03001-t001:** Characteristics and outcomes of pancreatic cancer-related malignant distal biliary obstruction patients with fully covered self-expandable metal stents.

Variable	PMS (*n* = 103)	PPS (*n* = 56)	*p*
Sex, males:females, *n*:*n*	68:35	34:22	0.603
Age, years, median (range)	67 (31–90)	69 (41–98)	0.090
TNM stage, II/III:IV, *n*:*n*	48:55	28:28	0.741
Tumor location, head/uncinate:body, *n*:*n*	99:4	52:4	0.453
Tumor size, mm, median (range)	31.5 (10–100)	30 (19–41)	0.724
Liver metastasis, *n* (%)	34 (33.01%)	16 (27.11%)	0.596
Ascites, *n* (%)	12 (11.65%)	5 (8.93%)	0.789
Total bilirubin, mg/dL, median (range)	7.5 (0.3–37.1)	3.1 (0.2–24.0)	<0.001
AST, mg/dL, median (range)	159 (8–939)	65 (5–412)	<0.001
ALP, mg/dL, median (range)	354 (29–2175)	255 (39–1030)	0.012
CEA, ng/mL, median (range)	4.21 (0.63–331)	4.76 (0.78–453)	0.714
CA-199, U/mL, median (range)	733 (0.6–50,000)	233.5 (0.3–50,000)	0.388
Duodenal invasion, *n* (%)	31 (30.09%)	22 (37.29%)	0.292
Cholangitis before SEMS, *n* (%)	8 (7.76%)	27 (48.21%)	<0.001
Stricture length, mm, median (range)	22 (10–40)	20 (10–40)	0.900
SEMS length, 6:7/8 cm, *n*:*n*	83:20	48:8	0.516
Complication, *n* (%)	11 (10.67%)	4 (7.14%)	0.577
Cholecystitis, *n* (%)	8 (7.76%)	2 (3.57%)	0.476
Pancreatitis, *n* (%)	3 (2.91%)	2 (3.57%)	1.000
Recurrent biliary obstruction, *n* (%)	22 (21.35%)	18 (32.14%)	0.180
Sludge/food impaction, *n* (%)	9 (40.91%)	10 (55.56%)	1.000
Tumor ingrowth, *n* (%)	3 (13.63%)	2 (11.11%)	1.000
Tumor overgrowth, *n* (%)	7 (31.82%)	4 (22.22%)	0.724
Migration, *n* (%)	3 (13.63%)	2 (11.11%)	1.000
Radiotherapy, *n* (%)	21 (20.38%)	18 (32.14%)	0.123
Chemotherapy, *n* (%)	59 (57.28%)	37 (66.07%)	0.395
SEMS patency duration, days, median (95% CI)	152 (132–213)	177.5 (145–228)	0.529

Abbreviations: PMS, primary metal stent; PPS, prior plastic stent; TNM, tumor/lymph node/metastasis; AST, aspartate aminotransferase; ALT, alanine aminotransferase; ALP, alkaline phosphatase; SEMS, self-expandable metal stent; CEA, carcinoembryonic antigen; CA-199, carbohydrate antigen-199; CI, confidence interval.

**Table 2 cancers-15-03001-t002:** Risk factors for recurrent biliary obstruction in fully covered self-expandable metal stents.

Variable	Univariable Analysis			Multivariable Analysis		
	HR	*p*-Value	95% CI	HR	*p*-Value	95% CI
Prior plastic stent drainage	1.131	0.464	0.814–1.570			
Bilirubin > 5 mg/dL	0.828	0.241	0.603–1.136	0.870	0.393	0.631–1.198
Cholangitis before SEMS	0.947	0.778	0.649–1.382			
Tumor size > 35 mm	1.309	0.814	0.755–1.430			
Stricture > 25 mm	1.086	0.609	0.794–1.488			
Duodenal invasion	0.897	0.520	0.643–1.520			
Ascites	0.689	0.150	0.416–1.143	0.748	0.267	0.448–1.249
Liver metastasis	0.849	0.342	0.605–1.190			
SEMS length > 6 cm	0.602	0.017	0.397–0.913	0.631	0.032	0.414–0.691

Abbreviations: CI, confidence interval; SEMS, self-expandable metal stent.

## Data Availability

The data presented in this study are available on request from the corresponding author. The data are not publicly available because of institutional restrictions.

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
