# Peer review of "Efficacy of Fully Covered Self-Expandable Metal Stents for Distal Biliary Obstruction Caused by Pancreatic Ductal Adenocarcinoma: Primary Metal Stent vs. Metal Stent following Plastic Stent"

_cancers, 2023, doi:10.3390/cancers15113001_

Round 1

Reviewer 1 Report

A concise learning study , easy to read and understand.

Author Response

Response to Reviewer 1 Comments

Author:

Thank you for your kind comments and recommendations.

Reviewer 2 Report

The authors are aware of the limitations (they state three). The anti-cancer treatment is especially important.

The subject matter is certainly interesting.

Author Response

Response to Reviewer 2 Comments

Thank you for your kind comments and recommendations. Further prospective studies are required to identify the effect of anti-cancer therapy, especially for different regimens of chemotherapy.

Reviewer 3 Report

This manuscript is an original article that retrospectively investigated the efficacy of fully covered self-expandable metal stent (FCSEMS) for distal biliary obstruction caused by pancreatic ductal adenocarcinoma, by comparing primary metal stent and metal stent following plastic stent. The authors showed that the recurrent biliary obstruction (RBO) rates and SEMS patency duration did not differ between the two groups. Furthermore, the authors identified that an FCSEMS greater than 6 cm in length was a risk factor for RBO using the multivariable analysis.

This study was conducted well, the methods are appropriate, and the data are presented clearly.

However, I have serious concern in this manuscript.

Major

1.     The characteristics of two groups are too heterogeneous to compare, especially in cholangitis rates before stenting which are the crucial point in this study. The authors should focus on the patients with cholangitis and separately analyze them. A prospective comparison would be desirable.

Minor

1. The authors selected 7-8 cm FCSEMSs in some patients with stricture length of only 20-40mm. I can’t understand why longer stents were indicated.

2.     (P1L33) “Does” should be deleted.

3.     Please describe the timing of stent-exchange from plastic to metallic ones.

4.     (P3L122-123) The authors stated that no anti-cancer treatments were administered, including chemotherapy or radiotherapy. However, there are some patients with chemotherapy or radiotherapy in Table 1. It’s confusing.

Author Response

Response to Reviewer 3 Comments

Point 1: The characteristics of two groups are too heterogeneous to compare, especially in cholangitis rates before stenting which are the crucial point in this study. The authors should focus on the patients with cholangitis and separately analyze them. A prospective comparison would be desirable.

Response 1: Thank you for your kind comments and recommendations. Cholangitis is one of the most important factors in our study. Cholangitis before FCSEMS placement was much higher in the prior plastic stent (PPS) group; however, the median days to recurrent biliary obstruction (RBO) was not significantly different between the primary metal stent (PMS) and PPS groups. To confirm the influence of cholangitis, we inserted cholangitis before metal stent as one of the factors in our Cox regression and revealed that cholangitis was not a risk factor for RBO.

Point 2: The authors selected 7-8 cm FCSEMSs in some patients with stricture length of only 20-40mm. I can’t understand why longer stents were indicated.

Response 2: Thank you for this comment. The length of the SEMSs that were to be placed was determined by the estimated obstruction length and was at least 10–20 mm longer than the sides of the bile duct stricture. Furthermore, the distal end of the metal stent was placed in the duodenal lumen and protruded from the papilla by approximately 1 cm. A long metal stent also had the benefit of less axial force to allow the metal stent to fit the bile duct wall.

Point 3: (P1L33) “Does” should be deleted.

Response 3: Thank you for pointing this out. We have adjusted the manuscript accordingly.

Point 4: Please describe the timing of stent-exchange from plastic to metallic ones.

Response 4: Thank you for this comment. We have added this detail within the manuscript as per your suggestion. Please see the “Materials and Methods” section.

Point 5: (P3L122-123) The authors stated that no anti-cancer treatments were administered, including chemotherapy or radiotherapy. However, there are some patients with chemotherapy or radiotherapy in Table 1. It’s confusing.

Response 5: Thank you for this comment. We have clarified within the manuscript that there was no difference between the two groups in post-stenting anti-cancer treatment, including chemotherapy or radiotherapy.

Round 2

Reviewer 3 Report

Thank you for revising the manuscript according to my suggestion. The revised manuscript is improved enough to be accepted.